# Prioritizing Image-Related Tokens Enhances Vision-Language Pre-Training

**Yangyi Chen**                                                                      *yangyic3@illinois.edu*
*University of Illinois Urbana-Champaign*

**Hao Peng**                                                                         *haopeng@illinois.edu*
*University of Illinois Urbana-Champaign*

**Tong Zhang**                                                                       *tozhang@illinois.edu*
*University of Illinois Urbana-Champaign*

**Heng Ji**                                                                          *hengji@illinois.edu*
*University of Illinois Urbana-Champaign*

**Reviewed on OpenReview:** *https://openreview.net/forum?id=jDcnL1hB1Z*

## Abstract

In standard large vision-language models (LVLMs) pre-training, the model typically maximizes the joint probability of the caption conditioned on the image via next-token prediction (NTP); however, since only a small subset of caption tokens directly relates to the visual content, this naive NTP unintentionally fits the model to noise and increases the risk of hallucination. We present PRIOR, a simple vision-language pre-training approach that addresses this issue by **prior**itizing image-related tokens through differential weighting in the NTP loss, drawing from the importance sampling framework. PRIOR introduces a reference model—a text-only large language model (LLM) trained on the captions without image inputs, to weight each token based on its probability for LVLMs training. Intuitively, tokens that are directly related to the visual inputs are harder to predict without the image and thus receive lower probabilities from the text-only reference LLM. During training, we implement a token-specific re-weighting term based on the importance scores to adjust each token's loss. We implement PRIOR in two distinct settings: LVLMs with visual encoders and LVLMs without visual encoders. We observe 19% and 8% average relative improvement, respectively, on several vision-language benchmarks compared to NTP. In addition, PRIOR exhibits superior scaling properties, as demonstrated by significantly higher scaling coefficients, indicating greater potential for performance gains compared to NTP given increasing compute and data. The code is available at https://github.com/Yangyi-Chen/PRIOR.

## 1 Introduction

Vision-language pre-training enhances both visual perception and visual-textual association capabilities in large vision-language models (LVLMs) (Zhang et al., 2021; Wu et al., 2024). However, our preliminary human annotations on 100 examples from Capsfusion (Yu et al., 2024) reveal that only 31.3% of words in web-scale image-caption pairs directly relate to the associated images, with the remainder containing irrelevant information, stylistic elements, or website-specific content. For example, considering the second example in Fig. 1 (Top), only "house", "front yard", and "lawn" are image-related, while the remaining tokens lack visual correspondence, such as the location and price information of the house. The standard next-token prediction (NTP) objective treats all tokens equally, regardless of their relevance to visual content. Thus, besides learning visual perception and vision-language alignment, LVLMs inevitably model the entire

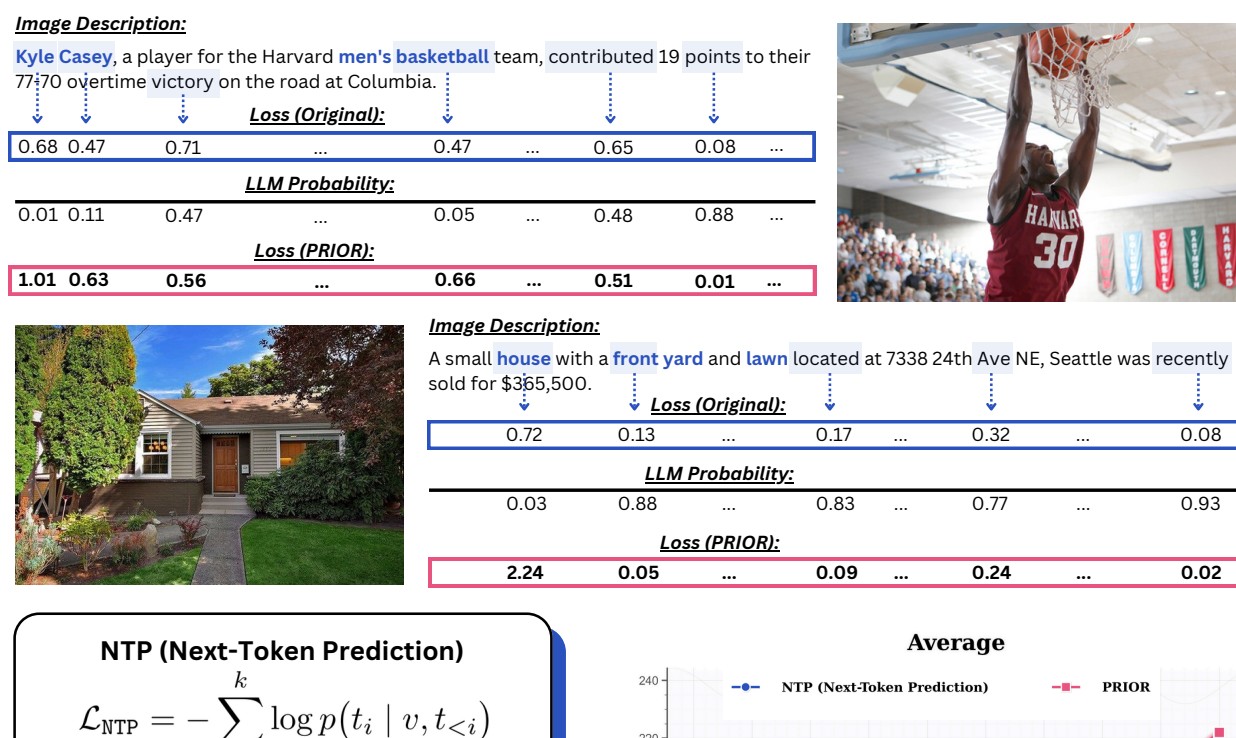

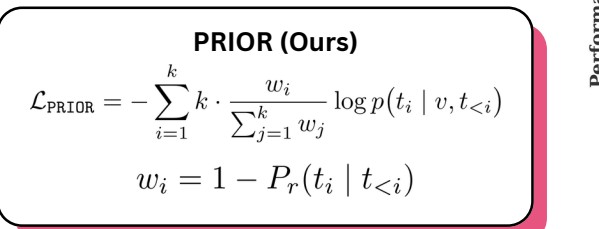

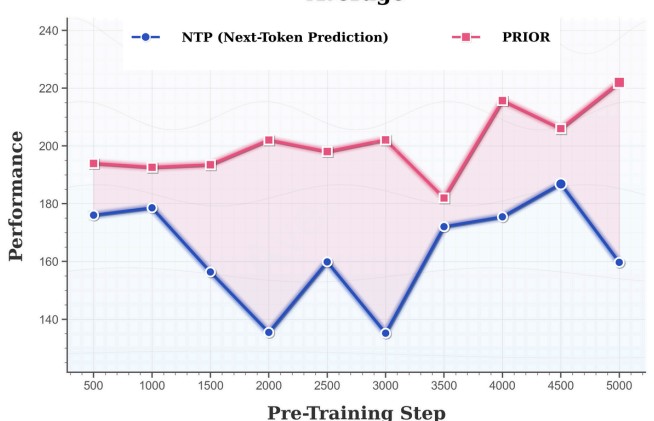

Figure 1: **(Top) Synthetic examples to highlight the motivation of PRIOR.** Only a few tokens in the captions (highlighted in blue, word-level for better visualization) are related to the associated images. PRIOR utilizes probability scores from a text-only LLM to recalibrate the original loss function at the token level, prioritizing image-related tokens that receive lower probability scores from the LLM. **(Bottom Left) PRIOR formulation.** Given an image $v$ paired with a caption '$t_1, t_2, ..., t_k$', PRIOR enhances vision-language pre-training by assigning a normalized weight to each token loss, which is computed based on the LLM probability $P_r(t_i|t_{<i})$. **(Bottom Right) Performance of PRIOR.** PRIOR demonstrates consistent performance improvement (average over several vision-language benchmarks) and better training stability compared to the widely used next-token prediction objective in vision-language pre-training. In addition, PRIOR shows superior scaling behaviors in both performance predictability and potential improvement with increased compute and data (§4).

caption distribution, potentially overfitting to noise and contributing to hallucination (Sharma et al., 2018; Changpinyo et al., 2021; Liu et al., 2023b).

In this work, we present **PRIOR**, a simple approach that enhances vision-language pre-training by **prior**itizing optimization on image-related tokens. PRIOR addresses the above challenge by effectively distinguishing between image-related tokens and other tokens in the training corpus. We train a text-only reference model to capture the caption distribution without visual context in the vision-language corpus. This allows us to identify which tokens are likely to be image-related and establish an importance distribution over the token

space. The key insight is that tokens easily predicted by the reference model likely contain minimal visual information, while those difficult for text-only models to predict more likely convey image-specific content. We formalize this intuition by using the text-only model's predictive probability to reweight the NTP loss for each token, as shown in Fig. 1 (Bottom left). This training algorithm, inspired by importance sampling principles, prioritizes the optimization on tokens containing image-related information while preserving general language generation capabilities.

PRIOR addresses several crucial limitations in previous vision-language pre-training approaches. Existing methods, as detailed in §6, primarily fall into two categories, each with their own challenges. The first category involves supplementing NTP with additional training objectives such as contrastive loss (Li et al., 2019) or distillation loss (Liao et al., 2025), but these approaches sacrifice NTP's simplicity and create challenges for large-scale, efficient implementation with existing frameworks (Rasley et al., 2020). In contrast, PRIOR can be easily integrated with existing pre-training frameworks through minimal code changes for the loss computation and offline-computed token importance scores. The second category focuses on optimizing training data through distillation (Chen et al., 2024a), filtering pipelines (Guo et al., 2024), and related approaches. While conceptually promising, these methods face inherent scalability constraints due to their dependence on teacher models or human-designed heuristics. In contrast, PRIOR's text-only reference model offers a scalable alternative that dynamically extracts useful knowledge from noisy web-scale datasets, ensuring broad compatibility with advancements across vision-language datasets.

To evaluate PRIOR's broad applicability across different LVLMs architectures, we conduct experiments with both vision encoder-based LVLMs (Zhu et al., 2023; Li et al., 2023b) and end-to-end unified LVLMs with a simple linear projector for image processing (Chen et al., 2024f; Diao et al., 2024; Tao et al., 2024). Our results demonstrate that PRIOR consistently outperforms the naive NTP pre-training baseline, achieving 19% and 8% average relative improvement across these two architectures respectively, while also enhancing training stability. The average performance comparisons are visualized in Fig. 1 (Bottom Right) and Fig. 4. In §3.4, we compare PRIOR with additional baselines and alternative token selection methods, and show the unique advantages of PRIOR. Further analysis in §4 reveals that PRIOR exhibits superior scaling behaviors in terms of performance predictability (Fig. 6) and potential improvement with increased computational resources (Fig. 7). Moreover, as detailed in §5.2, PRIOR accelerates pre-training by more effectively optimizing loss on both image-related and image-unrelated tokens.

## 2 PRIOR

### 2.1 Problem Formulation: Vision-Language Pre-training with Image-Caption Pairs

We review the general vision-language pre-training that leverages a web-scale dataset comprising image-caption pairs $(v, c)$ to develop models capable of generating relevant textual descriptions based on visual inputs. $v$ represents the image, and $c = t_1, t_2, ..., t_k$ represents the corresponding caption consisting of $k$ tokens. The widely used vision-language pre-training objective, next-token prediction ($\mathcal{L}_{\texttt{NTP}}$), trains the model to predict each token in the caption based on the image and all previous tokens (Liu et al., 2023d; Dai et al., 2023; Lu et al., 2024), formally expressed as:

$$\mathcal{L}_{\texttt{NTP}} = -\sum_{i=1}^{k} \log p(t_i \mid v, t_{<i}), \tag{1}$$

where $t_i$ is the current token to predict, $t_{<i}$ represents the prefix.

### 2.2 PRIOR: Prioritizing Image-Related Tokens for Vision-Language Pre-Training

Eq. 1 outlines a model that conditions token prediction on both images and text but lacks a mechanism to verify whether visual information is actually being effectively utilized. The model can optimize this objective by relying exclusively on textual context, potentially resulting in LVLMs that overfit to supplementary text that doesn't correspond to visible image content. To address this, we introduce PRIOR, a simple method to advance the original vision-language pre-training by prioritizing image-related tokens, which are

automatically identified by a text-only reference LLM. Drawing from the importance sampling framework (§2.3), we use the reference model to construct a target distribution that assigns higher probability to tokens likely requiring visual information. This approach effectively concentrates optimization on image-related content while reducing emphasis on tokens predictable from text alone.

**Text-Only LLM for Modeling Text Distribution**  PRIOR introduces a text-only LLM as a reference to model the caption distribution without the image inputs. Specifically, we apply NTP loss exclusively on the text tokens, and the training objective is:

$$\mathcal{L}_{\texttt{TEXT}} = -\sum_{i=1}^{k} \log p_r\big(t_i \mid t_{<i}\big). \tag{2}$$

**Vision-Language Pre-Training with Reweighted Tokens Loss**  PRIOR utilizes the reference model's token probability to calculate the **importance score** $w_i$ for each token:

$$w_i = \big(1 - p_r(t_i \mid t_{<i})\big)^{\alpha}, \tag{3}$$

where $\alpha$ (empirically set to 1) modulates the impact of $w_i$ during pre-training. Tokens with higher importance scores are those that the text-only LLM finds difficult to predict, suggesting they are more likely to be image-related. We implement this scoring process offline—computing and storing the importance score for each token beforehand—which eliminates the need for reference model inference during vision-language pre-training.

PRIOR applies these token-specific importance scores to reweight the NTP loss. The training objective is:

$$\mathcal{L}_{\texttt{PRIOR}} = -\sum_{i=1}^{k} k \cdot \frac{w_i}{\sum_{j=1}^{k} w_j} \log p\big(t_i \mid v, t_{<i}\big) \tag{4}$$

The normalization term $w_i / \sum_{j=1}^{k} w_j$ ensures that the importance scores form a proper distribution across all $k$ tokens, while the multiplication by $k$ preserves the overall scale of the loss. This algorithm strategically upweights tokens that the text-only model struggles to predict, which typically correspond to image-specific information, while maintaining sufficient weight distribution across contextual tokens to ensure coherent language generation.

### 2.3  PRIOR as Importance Sampling

We present the theoretical framework based on importance sampling that underpins and motivates PRIOR. In general, PRIOR can be interpreted as a form of importance sampling where we draw more samples from regions where the reference model is uncertain. In classical importance sampling, we estimate an expectation under a target distribution $p(x)$ using samples from a proposal distribution $q(x)$:

$$\mathbb{E}_{p(x)}[f(x)] = \int f(x)p(x)dx = \int f(x)\frac{p(x)}{q(x)}q(x)dx = \mathbb{E}_{q(x)}\left[f(x)\frac{p(x)}{q(x)}\right], \tag{5}$$

where the term $\frac{p(x)}{q(x)}$ represents the importance weight that corrects for the mismatch between the distributions. For PRIOR, we can formulate our problem as follows:

- **Function** $f(x)$: The NTP loss (*a.k.a,* negative log-likelihood loss) $-\log p_{\texttt{model}}(t_i|v, t_{<i})$.
- **Sampling Distribution** $q(x)$: The empirical distribution in data $p_{\texttt{data}}(t_i|v, t_{<i})$.
- **Target Distribution** $p(x)$: We aim to define a new target distribution $p_{\texttt{target}}(t_i|v, t_{<i})$ that assigns higher probability to tokens that are difficult for the reference model to predict:

$$p_{\texttt{target}}(t_i|v, t_{<i}) \propto p_{\texttt{data}}(t_i|v, t_{<i}) \cdot \big(1 - p_{\text{r}}(t_i|t_{<i})\big), \tag{6}$$

where $p_{\text{r}}(t_i|t_{<i})$ is the reference model's probability estimate. Intuitively, we upweight tokens that have low reference probability (surprising tokens).

**Importance Sampling Formulation**: Since we only have samples from $p_{\texttt{data}}(t_i|v, t_{<i})$, we use importance sampling to estimate the expectation under $p_{\texttt{target}}(t_i|v, t_{<i})$:

$$\mathbb{E}_{p_{\texttt{target}}(t_i|v,t_{<i})}[-\log p_{\texttt{model}}(t_i|v, t_{<i})] =$$

$$\mathbb{E}_{p_{\texttt{data}}(t_i|i,t_{<i})}\left[-\log p_{\texttt{model}}(t_i|v, t_{<i}) \cdot \frac{p_{\texttt{target}}(t_i|v, t_{<i})}{p_{\texttt{data}}(t_i|v, t_{<i})}\right] \tag{7}$$

Thus, the importance weight for each token loss (*i.e.,* $-\log p_{\texttt{model}}(t_i|v, t_{<i})$ in Eq. 7), with the target distribution defined in Eq. 6, is:

$$w(t_i|v, t_{<i}) = \frac{p_{\texttt{target}}(t_i|v, t_{<i})}{p_{\texttt{data}}(t_i|v, t_{<i})} \propto \left(1 - p_{\text{r}}(t_i|t_{<i})\right) \tag{8}$$

To make this a proper probability distribution over the sequence, we normalize:

$$\tilde{w}(t_i|v, t_{<i}) = \frac{w(t_i|v, t_{<i})}{\sum_{j=1}^{k} w(t_j|v, t_{<j})} \tag{9}$$

While this self-normalization introduces bias into our estimation of the expectation over the target distribution, it substantially reduces variance, yielding improved training stability—a well-established tradeoff in previous work (Metelli et al., 2018; Korbak et al., 2022). Our importance-sampled loss thus becomes:

$$L_{\text{IS}} = -\sum_{i=1}^{k} \tilde{w}_t(t_i|v, t_{<i}) \cdot \log p_{\texttt{model}}(t_i|v, t_{<i}) \tag{10}$$

This weighted average represents the expected loss under our target distribution that emphasizes difficult tokens. Note that this formulation is consistent with Eq. 4, with the difference in the multiplication by $k$, which preserves the expected loss magnitude when transitioning from NTP to importance-weighted loss. Since the normalized weights sum to 1, omitting $k$ reduces the loss to a weighted average rather than a weighted sum, effectively shrinking the gradients by a factor of approximately $k$. Multiplying by $k$ restores the original scale. This design choice is important from two perspectives. First, it ensures **gradient consistency**: gradient magnitudes are preserved, enabling direct reuse of existing learning rates and optimizer configurations without retuning. Second, it provides **theoretical grounding** by aligning with self-normalized importance sampling, where normalization reduces variance at the cost of introducing a small bias (Metelli et al., 2018). In addition, this importance sampling framework can be easily extended to consider the $\alpha$ term. We also provide a theoretical justification for the key intuition of `PRIOR` through mutual information analysis, as detailed in §A.

## 3 Experiments

### 3.1 Implementation Details of `PRIOR`

We adopt CapsFusion (Yu et al., 2024), which comprises 120M image-text pairs for vision-language pre-training experiments. We first sample a subset (~5M) and use only the caption to pre-train a text-only reference LLM (initialized with Llama-3-8B (Dubey et al., 2024)). Since the captions in vision-language datasets are typically shorter, we pack multiple samples within the context length (8192) for efficient pre-training.

For vision-language pre-training, we sample another subset (~3M) and compute token-level reference probabilities $p_r(t_i|t_{<i})$ offline using the reference LLM. We store this data as (image, caption, token-level reference probability list) tuples. To measure the extensive applicability of `PRIOR` across diverse LVLMs architectures, we investigate two types of LVLMs:

- **H-LVLMs**: We implement LVLMs with pre-trained visual encoders (*i.e.,***H**eterogeneous architectures). Following the typical LLaVA-Style LVLMs design (Liu et al., 2023d), we adopt a straightforward ViT-MLP-LLM architecture, employing Llama-3.2-3B as the LLM backbone for efficient experimentation and CLIP-ViT-Large-336 (Radford et al., 2021) as the vision encoder. Following Liu et al. (2023c), we only train the MLP component during pre-training to learn the vision-language alignment and maintain consistent language capabilities across all LVLMs for our controlled study.

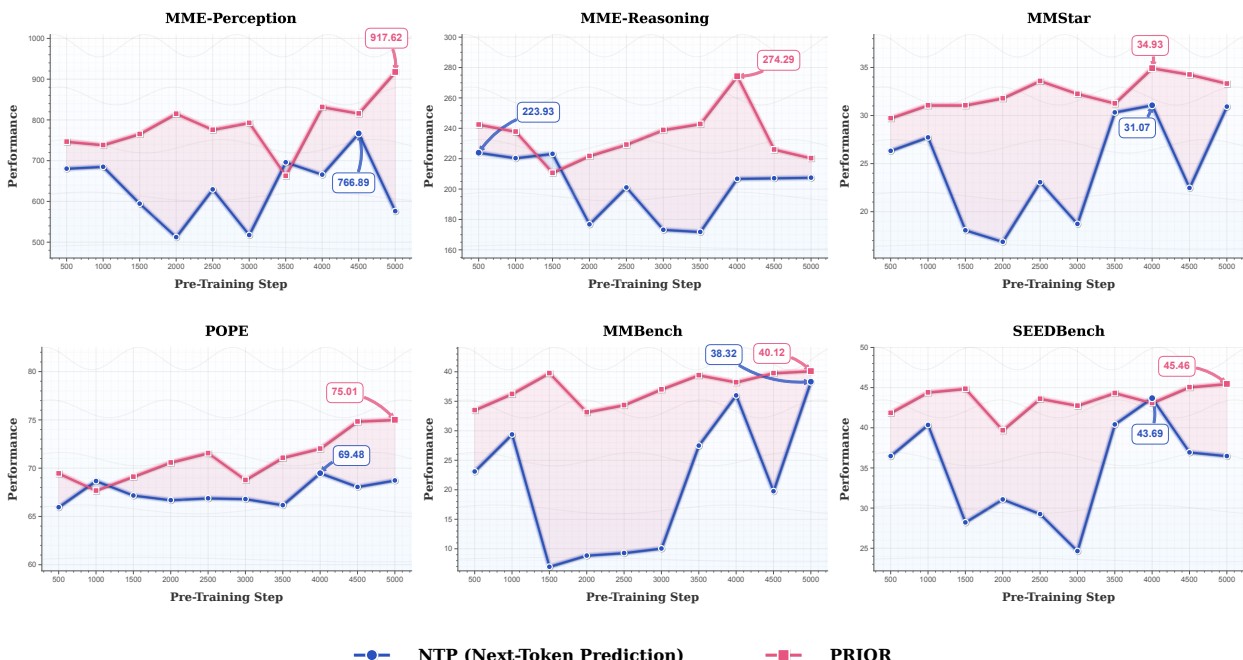

Figure 2: **Main experimental results of LVLMs with pre-trained visual encoders.** We compare PRIOR with the NTP vision-language pre-training across various training steps on LVLMs with pre-trained visual encoders, and we annotate the highest performance for each method, respectively. PRIOR demonstrates both superior performance and greater stability throughout the entire training.

- **U-LVLMs**: We implement LVLMs with **U**nified architectures, which drop the pre-trained visual encoders and adopt a single end-to-end Transformer architecture for vision-language modeling (Chen et al., 2024f; Lei et al., 2025; Zhang et al., 2025). Following Chen et al. (2024f), we initialize with Llama-8B and pre-train it on ImageNet for 500 steps before further pre-training on image-caption data.

For the two types of LVLMs, we set the total training steps as 5,000, record every 500 steps, with the batch size as 512. During U-LVLM pre-training, we also incorporate an equal proportion of DCLM language dataset (Li et al., 2024b) to maintain general language capabilities.

## 3.2 Experimental Setting

**Evaluation Benchmarks**  We select the following benchmarks for evaluation: (1) MME (Fu et al., 2024), which measures both the perception and reasoning capabilities. Thus, we report MME-Perception and MME-Reasoning respectively. (2) MMStar (Chen et al., 2024b), which measures advanced vision-language skills. (3) POPE (Li et al., 2023d), a benchmark for object hallucination evaluation. (4) MMBench (Liu et al., 2024c), a benchmark designed for general vision-language capacities evaluation. (5) SEEDBench (Li et al., 2024a), which covers 12 evaluation dimensions including diverse aspects of LVLMs.

**Evaluation Setting**  To elicit meaningful responses beyond image captions during evaluation, we conduct 20 steps for instruction fine-tuning on all pre-trained checkpoints using the same data subset from the post-training set in Chen et al. (2024f). We include a controlled sensitivity study over the number of fine-tuning Steps in §B. We employ VLMEvalKit (Duan et al., 2024) for unified evaluation and comparison across models.

## 3.3 Experimental Results

The main experimental results for each dataset are shown in Fig. 2 (H-LVLMs) and Fig. 3 (U-LVLMs). Average performance across all datasets is summarized in Fig. 1, bottom right (H-LVLMs) and Fig. 4 (U-LVLMs). We have the following findings:

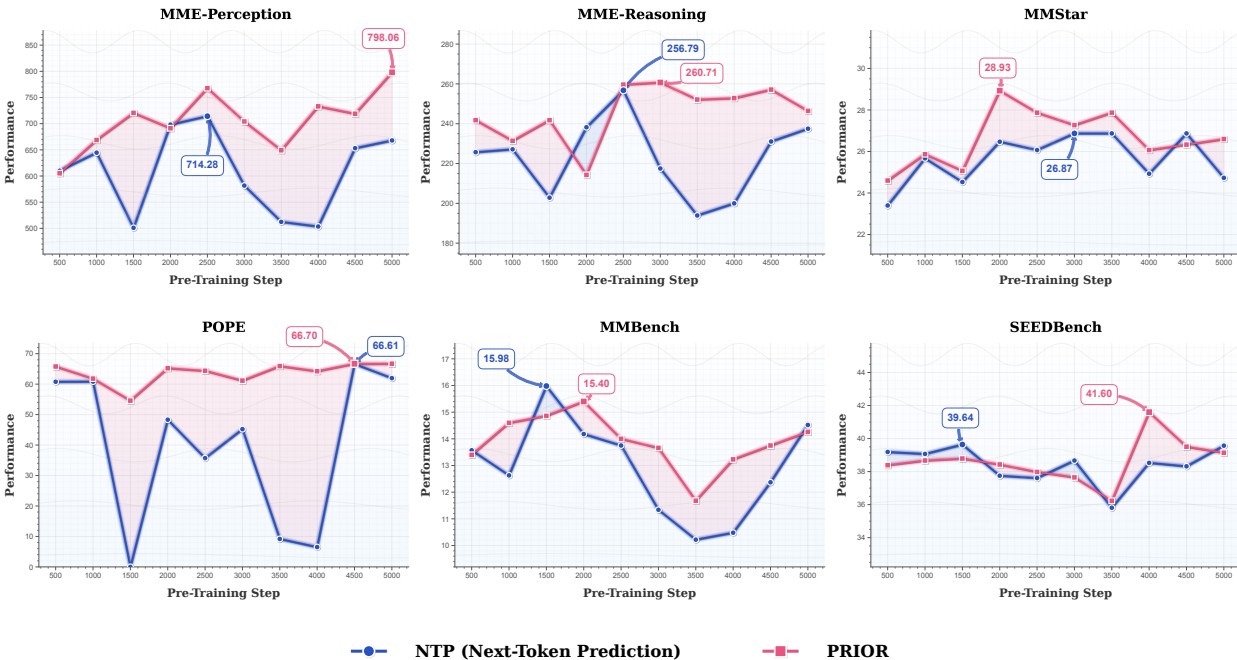

Figure 3: **Main experimental results of LVLMs with unified architectures.** We compare `PRIOR` with the NTP vision-language pre-training on LVLMs with unified architectures.

- **`PRIOR` generally outperforms the naive NTP pre-training objective by a large margin.** We observe that `PRIOR` enhances performance across the entire training trajectory for both H-LVLMs and U-LVLMs, demonstrating improvements compared to the NTP objective at most intermediate training steps. For fully converged models (*i.e.,* at 5,000 steps), we observe average relative improvements across all datasets of 18.61% for H-LVLMs and 7.93% for U-LVLMs.

- **`PRIOR` demonstrates better training stability.** We observe performance fluctuations during training across both H-LVLMs and U-LVLMs. The NTP objective causes significant performance drops in MMBench and SEEDBench for H-LVLMs, while producing sharp performance fluctuations in POPE and MMBench for U-LVLMs. `PRIOR` significantly reduces these fluctuations, demonstrating superior training stability compared to the naive NTP pre-training objective.

- **`PRIOR` exhibits higher potential in performance.** The highest performance scores are highlighted in Fig. 2 and Fig. 3 for each dataset. Our results show `PRIOR` outperforms the NTP objective in peak performance for both LVLM types. For H-LVLMs specifically, `PRIOR` achieves optimal performance in final checkpoints, while NTP peaks earlier in training. This suggests `PRIOR` benefits from extended training, likely due to its comprehensive optimization approach that continues refining multimodal representations in later stages. `PRIOR`'s sustained improvement trajectory indicates greater performance potential.

### 3.4 Comparison with Additional Baselines, Token Selection Methods, and Selected Reference LLMs

We compare `PRIOR` with the following approaches specifically designed for H-LVLMs: (1) **Image-text matching**: The training objective that pre-trains the MLP connector to map the [`CLS`] token embedding close to the caption embedding (Li et al., 2023b; Chen et al., 2024g). (2) **Image-text contrastive learning**: The training objective that pre-trains the MLP connector to align projected image representations with text representations (Chen et al., 2024g; Radford et al., 2021). (3) **Reconstructive tuning**: The training objective that supervises LVLMs to reconstruct images, focusing on the inherent richness and detail within input images (Wang et al., 2024). All these methods are combined with NTP for 5,000 training steps.

Table 1: **A comparison of `PRIOR` with additional baselines and alternative token-selection approaches implemented on H-LVLMs.** The evaluation is the average of two runs. `PRIOR` yields consistently better results throughout the evaluation.

| Category | Method | MME-P | MME-R | MMStar | POPE | MMBench | SEEDBench |
|---|---|---|---|---|---|---|---|
| **Naive** | **Next-Token Prediction** | 576.2 | 207.5 | 30.9 | 68.7 | 38.3 | 36.5 |
| **Baseline** | **Image-Text Matching** | 580.3 | 215.0 | 30.8 | 68.9 | 38.1 | 42.5 |
| | **Image-Text Contrastive** | 635.6 | 218.6 | 26.3 | 67.5 | 38.2 | 41.2 |
| | **Reconstructive Tuning** | 648.3 | 202.1 | 27.3 | 69.3 | 38.0 | 41.5 |
| **Token Selection** | **Attention Weighting** | 655.4 | 215.8 | 31.2 | 73.4 | 36.9 | 39.2 |
| | **Rare Word Weighting** | 588.1 | 203.2 | 30.1 | 70.4 | 39.2 | 38.5 |
| **Selected Reference LLMs** | **PRIOR (8B off-the-shelf LLM)** | 703.4 | 203.6 | 29.3 | 68.6 | 31.4 | 41.6 |
| | **PRIOR (3B fine-tuned LLM)** | 824.2 | 222.9 | 32.1 | 71.0 | 38.0 | 44.6 |
| | **PRIOR (On-the-fly inference)** | 856.0 | 210.7 | 31.7 | 67.0 | 38.3 | 42.5 |
| | **PRIOR (8B fine-tuned LLM)** | **917.6** | **220.4** | **33.3** | **75.0** | **40.1** | **45.5** |

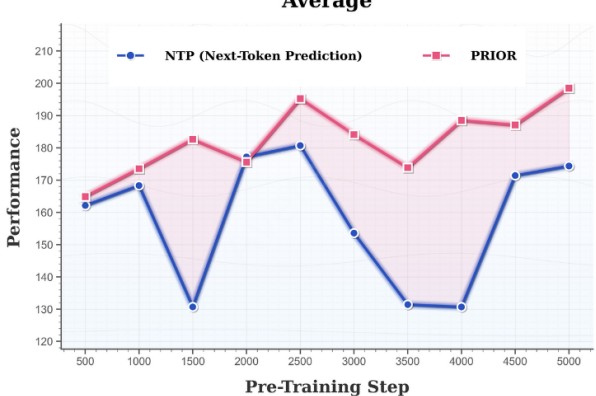

Figure 4: **The average performance comparison on LVLMs with unified architectures.** `PRIOR` demonstrates better performance and stability across the entire training process.

Figure 5: **The ablation study of `PRIOR` regarding $\alpha$ and $k$ on H-LVLMs.** Results show optimal performance with $\alpha=1$ and scaling factor $k$, justifying the design choices in `PRIOR`.

The results presented in Tab. 1 demonstrate that `PRIOR` consistently outperforms these baselines across all benchmarks, with particularly significant gains observed on complex reasoning tasks (*i.e.,* MME-R, MMStar). Additionally, `PRIOR` integrates more seamlessly with existing pre-training frameworks, requiring only modifications to the loss computation when data is stored offline. This enables scalable pre-training of LVLMs on large-scale datasets across compute clusters.

To further validate the effectiveness of our approach, we conduct another study comparing `PRIOR` against alternative token-weighting strategies. Since `PRIOR` operates by prioritizing optimization on *important* tokens within LVLMs, we examine two additional heuristic methods for computing token importance scores: (1) **Attention weighting**, which leverages cross-attention scores from a pre-trained CLIP model (Radford et al., 2021) to assign weights to text tokens based on their visual relevance. (2) **Rare word weighting**, which computes importance scores using inverse token frequency statistics from large-scale text corpora (Das et al., 2023), under the assumption that rare tokens carry more semantic significance.

As shown in Tab. 1, `PRIOR` substantially outperforms these two token-weighting methods across all evaluation benchmarks, validating the superiority of our principled approach to token importance estimation, with gains of +262.2 and +329.5 on MME-P, a critical benchmark that examines the visual perception capabilities of LVLMs, respectively. This substantial margin highlights that our importance sampling formulation of token importance, grounded in the `PRIOR` objective, more effectively identifies semantically critical tokens than attention-based or frequency-based heuristics.

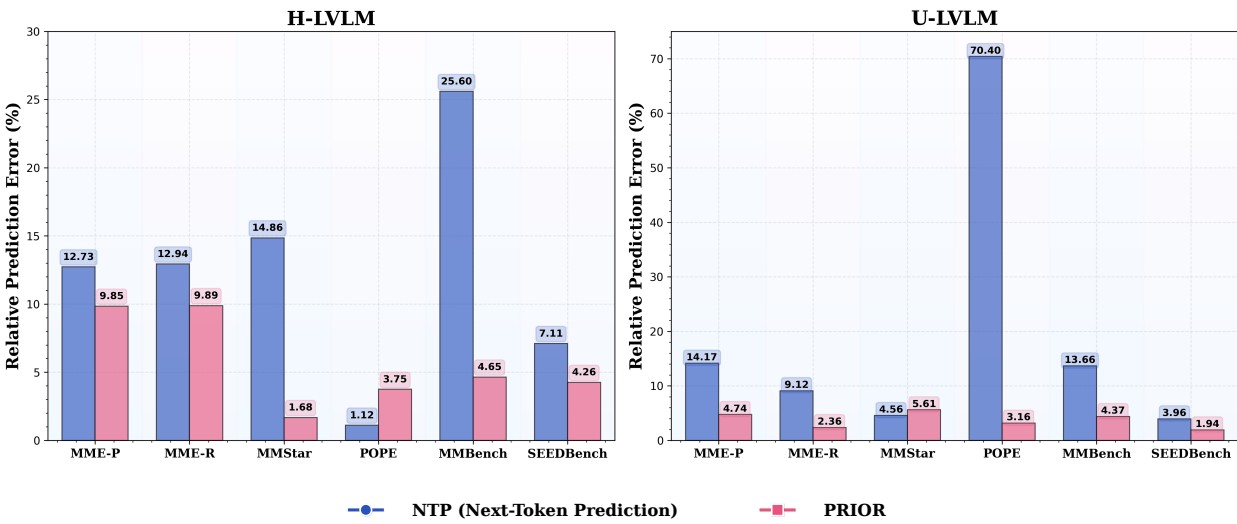

Figure 6: **The relative prediction error of NTP and PRIOR.** We observe that the performance of LVLMs trained via PRIOR is more predictable at scale.

We further conduct ablation studies to examine the effect of reference LLM configuration (*i.e.,* fine-tuned vs. off-the-shelf, 3B vs. 8B) as well as an on-the-fly inference alternative, with results on H-LVLMs after 5,000 pretraining steps reported in Tab. 1. We observe that using an off-the-shelf 8B LLM without fine-tuning leads to substantial degradation across all benchmarks (*e.g.,* MME-Perception drops from 917.6 to 703.4), demonstrating that task-specific adaptation is critical for generating reliable importance scores. Among fine-tuned variants, scaling the reference model from 3B to 8B yields consistent improvements, with notable gains on MME-Perception (+93.4) and POPE (+4.0). Nonetheless, even the smaller 3B fine-tuned model provides meaningful guidance for larger LVLMs, indicating that the proposed framework is robust to scale mismatch between the reference and target models, though a scale-matched configuration remains preferable. We also compare against an on-the-fly baseline that estimates text-only predictability during training by running a caption-only forward pass (with images masked) to compute importance weights, followed by an image+caption forward pass optimizing the reweighted loss. PRIOR with offline precomputed scores from the 8B fine-tuned LLM substantially outperforms this approach across all benchmarks, with notable improvements on MME-Perception (+61.6), POPE (+8.0), and SEEDBench (+3.0). The reason is that the caption-domain fine-tuned reference LLM is specifically optimized for modeling caption distributions, yielding more reliable estimates of text-only predictability than the LVLM's masked forward pass.

## 3.5 Ablation Study

We conduct an ablation study to understand the influence of $\alpha$ and $k$ in Eq (4). While PRIOR sets $\alpha$ to 1 by default, we experiment with $\alpha = 0.5, 2, 4$ to test sensitivity. Higher $\alpha$ increases focus on optimizing loss for image-related tokens identified by the reference model. Additionally, we evaluate the necessity of $k$, which maintains the token loss scale, by comparing performance with and without $k$ while fixing $\alpha = 1$. All experiments are conducted on H-LVLMs. Results in Fig. 5 demonstrate that performance degrades as $\alpha$ increases beyond 1 (to 2 or 4), yet remains stable within the moderate range of 0.5-1. Furthermore, removing $k$ negatively impacts performance on several benchmarks, validating our design choice to preserve the original loss scale. These findings suggest that a balanced approach to token weighting is crucial, as excessive emphasis on image-related tokens can impede the model's overall learning dynamics. Our empirical selection of $\alpha = 1$ and inclusion of scaling factor $k$ represents an optimal trade-off between focusing on visual content and maintaining stable training.

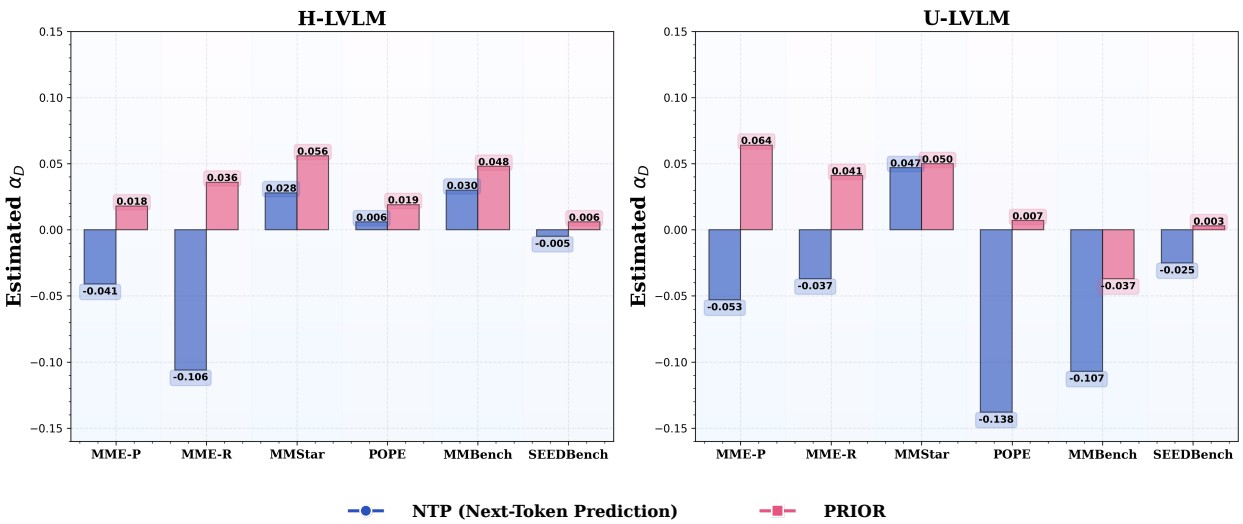

Figure 7: **The scaling behavior comparison of NTP and `PRIOR`.** `PRIOR` shows better scaling coefficients, indicating higher efficiency in translating increased resources into performance gains.

## 4 Scaling Laws: `PRIOR` Scales Predictably and Reliably with Increasing Compute

A fundamental distinction exists between LLMs and LVLMs regarding pre-training objectives. While NTP has proven to be an effective proxy for downstream task performance in LLMs (Chen et al., 2024e; Huang et al., 2024), this correlation does not extend to LVLMs (Chen et al., 2024f). Specifically, optimizing LVLMs to better predict caption given images through NTP does not reliably improve performance on downstream benchmarks. In this section, we conduct a critical analysis of the scaling behaviors of NTP and `PRIOR`. We train both H-LVLMs and U-LVLMs using different amounts of data (ranging from 7M to 70M training tokens, and including 8 sampling models), and use the following analytical form to estimate the scaling laws of NTP and `PRIOR` (Kaplan et al., 2020):

$$L(D) = \left( \frac{D}{D_c} \right)^{\alpha_D}, \tag{11}$$

where $\alpha_D$ and $D_c$ are constants to be estimated, $D$ is the amounts of token used in training, and $L$ is the predictive performance. This analytical form differs slightly from Kaplan et al. (2020) since we are targeting the benchmark performance rather than the pre-training loss. A larger $\alpha_D$ indicates superior scaling behavior, reflecting greater performance improvement for an equivalent amount of training tokens. With the fitted analytical function, we investigate two problems:

- **Predictability of model performance:** We use the fitted function to estimate the performance of LVLMs trained on 100M tokens, and measure the relative prediction error:

$$\text{Relative Prediction Error} = \frac{|\text{Predictive Performance} - \text{Actual Performance}|}{\text{Actual Performance}} \tag{12}$$

  The results are presented in Fig. 6. The downstream performance of both H-LVLMs and U-LVLMs exhibits significantly higher predictability when trained using `PRIOR`. This property facilitates more reliable performance estimation for larger-scale deployments, addressing a critical challenge in production-level LVLM implementation

- **Scaling behavior:** Fig. 7 illustrates the scaling factor (*i.e.,* $\alpha_D$) comparison for both methods. `PRIOR` demonstrates better scaling properties across two LVLMs architectures, consistently achieving higher $\alpha_D$ than NTP. This indicates that `PRIOR` more efficiently converts additional training data and compute into improved downstream performance.

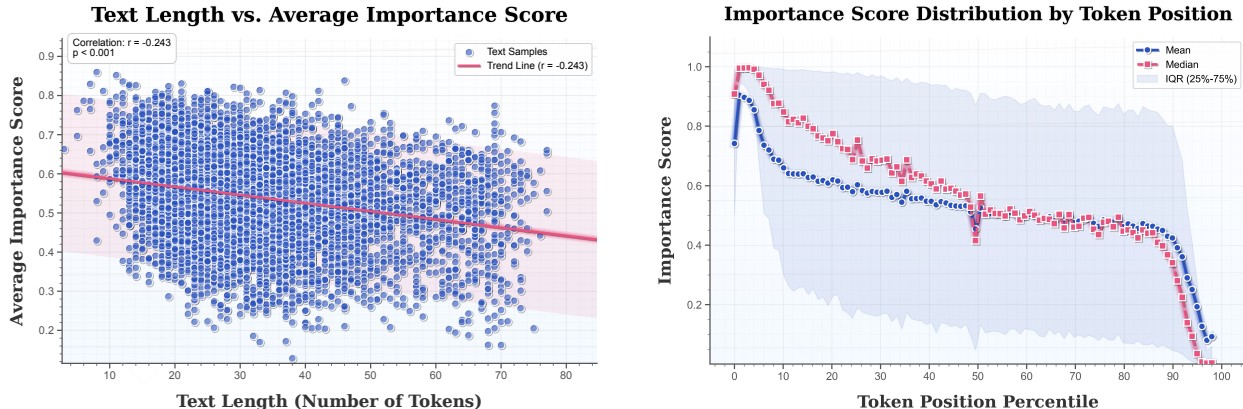

Figure 8: **Quantitative analysis of the importance score distribution.** We find that the average importance score for each caption decreases (linearly) with the text length. Within each caption, the importance score decreases in later positions.

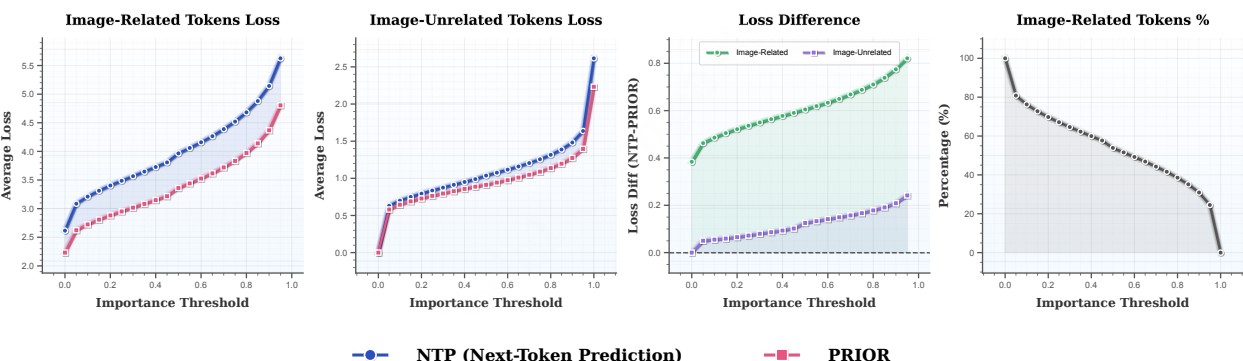

Figure 9: **The comparison of NTP and PRIOR regarding the achieved loss on image-related and image-unrelated tokens.** These two token groups are dynamically categorized based on varying the importance threshold $w_i$ (Eq. 3). We find that PRIOR accelerates the LVLMs training, consistently achieving lower loss on both image-related and image-unrelated tokens.

## 5 Further Analysis

### 5.1 Importance Score Distribution

PRIOR adjusts the loss for each token based on the importance score $w_i$ assigned to that token. We conduct a quantitative analysis to understand the distribution of the importance score. The results are presented in Fig. 8. We examine the relation between text length (*i.e.,* number of tokens) and the average importance score across all tokens within the caption, finding that the average importance score decreases as text length increases, with a linear correlation of $r$=-0.243. This suggests longer caption introduces more information absent from images, potentially including background content. Analysis of importance scores by token position reveals that initial tokens receive high importance scores, with scores progressively declining for later tokens. This indicates that later tokens can be more readily predicted from previous tokens without image reference.

### 5.2 Comparative Loss Analysis of NTP and PRIOR

We compare NTP and PRIOR from the loss perspective. By applying various importance thresholds, we categorize tokens into image-related and image-unrelated subsets based on their assigned importance scores $w_i$. We measure the average NTP loss of H-LVLMs trained via two methods on these two distinct subsets. The results in Fig. 9 reveal a significant pattern: PRIOR deliberately guides models to prioritize image-related tokens, achiev-

ing consistently lower loss on these information-rich elements. Interestingly, `PRIOR` also optimizes learning on image-unrelated tokens, though this performance difference is less pronounced than for image-related tokens. This suggests that `PRIOR` introduces general model improvement and effectively accelerates LVLMs training overall. Furthermore, our analysis reveals a compelling trend: as we progressively increase the importance threshold, the performance gap between methods widens substantially. This monotonic relationship confirms that `PRIOR` delivers increasingly significant improvements for tokens with higher image relevance, validating its fundamental design principle of prioritizing tokens that carry the most visually related information.

## 6  Related Work

### 6.1  Large Vision-Language Models

Existing research advances the development of LVLMs capable of addressing diverse tasks via a unified interface that can directly generate natural language, thus avoiding task-specific modifications (Wang et al., 2021; 2022; Li et al., 2023b; Agrawal et al., 2024; Luo et al., 2025). Utilizing advanced pre-trained LLMs (Brown et al., 2020; Bubeck et al., 2023; Dubey et al., 2024; Yang et al., 2024) as the language component (Liu et al., 2023d; Zhu et al., 2023), the instruction-following and complex reasoning abilities of LVLMs are significantly improved (Du et al., 2022; Ghosh et al., 2024). Typically, LVLMs leverage extensive image-caption pair datasets (Lin et al., 2014; Schuhmann et al., 2021; 2022) to train a projector that maps image features into the embedding space of LLMs, thereby aligning the two modalities (Liu et al., 2023d; Zhu et al., 2023; Alayrac et al., 2022; Li et al., 2023b; Yu et al., 2023; Awadalla et al., 2024). Furthermore, large-scale vision-language instruction tuning datasets (Su et al., 2023; Wei et al., 2023; Liu et al., 2023a; Gong et al., 2023; Gao et al., 2023; Li et al., 2023a; 2025) and feedback datasets (Chen et al., 2023a; Li et al., 2023c; Sun et al., 2023; Chou et al., 2024; Zhao et al., 2025) are utilized to align LVLMs with human preferences, ensuring their ability to comprehend instructions and generate responses that are user-friendly. In this work, we present a simple vision-language pre-training algorithm applicable to existing datasets that enhances visual-related training outcomes.

### 6.2  Vision-Language Training Objective

NTP loss on language tokens serves as the predominant pre-training objective for LVLMs (Wang et al., 2022; Dai et al., 2023; Bai et al., 2023; Lu et al., 2024; Dai et al., 2024; Chen et al., 2024f). Researchers have explored complementary approaches, including but not limited to: (1) Contrastive or matching loss to align the vision-language modalities (Li et al., 2019; Lu et al., 2019; Li et al., 2022). (2) Distillation loss to expedite the training (Diao et al., 2024; Liao et al., 2025). (3) Grounding objective to more effectively align the text tokens to the corresponding image regions (Koh et al., 2023; Rasheed et al., 2024). (4) Reconstructing the image based on the pre-defined codebook or external decoders (Sun et al., 2024; Zou et al., 2023; Ge et al., 2024). (5) External constrains to improve the visual perception in LVLMs (Wang et al., 2024; Luo et al., 2024a; Chen et al., 2023b). These supplementary objectives, though effective, compromise the simplicity of NTP in the original pre-training approach, complicating large-scale, efficient implementation (Shoeybi et al., 2019; Rasley et al., 2020; Liang et al., 2024; Zheng et al., 2023). `PRIOR` prioritizes maintaining NTP simplicity by merely adding a regularization term (*i.e.,* a weighting factor) to each token during training.

### 6.3  Vision-Language Training Corpus

The quality of training data is a critical determinant of LVLMs' ultimate performance. The established efforts on improving the training data include but not limited to: (1) Distilling knowledge from advanced closed-source LVLMs (Chen et al., 2024a;c; Liu et al., 2024b), like GPT-4o (Hurst et al., 2024). (2) Scaling the size of the dataset with scalable data collection pipelines (Li et al., 2024d; Awadalla et al., 2024; Chen et al., 2024d; Wang et al., 2025). (3) Incorporating the human-written filtering rules or pipelines (Guo et al., 2024; Gohari et al., 2025). (4) Curating domain-specific corpus to facilitate certain abilities in LVLMs (Li et al., 2024c; Yun et al., 2024; Fan et al., 2024; Han et al., 2024; Yang et al., 2025). (5) Relying on the self-evolving ability in LVLMs (Liu et al., 2024a; Luo et al., 2024b). While existing work makes valuable progress toward higher-quality training data, these approaches face fundamental scalability constraints imposed by their data creation components, including the inherent capabilities of teacher models, pre-defined

filtering criteria, and established pipelines—all of which create practical upper bounds on quality improvement. `PRIOR` avoids imposing human priors on the training dataset and remains compatible with advancements across all vision-language datasets.

# 7 Conclusion

This work introduces `PRIOR`, an advanced vision-language pre-training method that prioritizes the loss optimization on image-related tokens that a text-only reference model struggles to predict. Our experiments demonstrate that `PRIOR` significantly outperforms NTP, achieving 19% and 8% average relative improvement when implemented on LVLMs with and without pre-trained visual encoders, respectively. Furthermore, `PRIOR` exhibits more predictable and reliable scaling behaviors given increasing compute, indicating that `PRIOR` represents a promising pre-training algorithm.

## Limitations and Broader Impacts

**Limitations** `PRIOR` requires the LVLMs and the reference LLM to share the same tokenizer, preventing the creation of a universal dataset for all LVLMs pre-training. This constraint on the tokenizer necessitates specific implementations and processing, increasing computational overhead when deploying across multiple model architectures. However, this tokenizer constraint is a common practice in knowledge distillation and model training pipelines, and many modern model families (*e.g.,* LLaMA series (Dubey et al., 2024), Qwen series (Yang et al., 2024)) already share tokenizers across their variants of different sizes, making this limitation less restrictive in practice. We also discuss the limitations of `PRIOR`' token importance heuristic in §C.

**Broader impacts** `PRIOR`'s token prioritization approach enhances LVLMs' performance across benchmarks, potentially accelerating progress in building advanced general-purpose LVLMs. Additionally, by reducing hallucination risk through better grounding of language in visual content, `PRIOR` contributes to developing more trustworthy LVLMs models. However, there remains the possibility of misuse, and the differential weighting approach may unintentionally amplify existing biases in training data. We encourage continued research into responsible deployment practices and bias mitigation techniques alongside performance improvements.

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

Table 2: Sensitivity study over the number of SFT steps on H-LVLMs. We report performance of NTP and PRIOR along with their gap ($\Delta$) across six benchmarks. PRIOR consistently outperforms NTP regardless of the SFT duration, confirming that its gains stem from improved pre-training representations.

| SFT Steps | Method | MME-P | MME-R | MMStar | POPE | MMBench | SEEDBench |
|---|---|---|---|---|---|---|---|
| | NTP | 576.2 | 207.5 | 30.9 | 68.7 | 38.3 | 36.5 |
| Step-20 | PRIOR | **917.6** | **220.4** | **33.3** | **75.0** | **40.1** | **45.5** |
| | $\Delta$ | +341.4 | +12.9 | +2.4 | +6.3 | +1.8 | +9.0 |
| | NTP | 871.0 | 224.7 | 32.7 | 68.9 | 42.7 | 42.8 |
| Step-50 | PRIOR | **995.8** | **235.2** | **37.0** | **75.5** | **48.1** | **47.6** |
| | $\Delta$ | +124.8 | +10.5 | +4.3 | +6.6 | +5.4 | +4.8 |
| | NTP | 992.9 | 237.8 | 34.5 | 69.3 | 52.6 | 44.3 |
| Step-500 | PRIOR | **1024.7** | **243.9** | **37.3** | **76.3** | **59.7** | **50.4** |
| | $\Delta$ | +31.8 | +6.1 | +2.8 | +7.0 | +7.1 | +6.1 |
| | NTP | 947.0 | 241.6 | 37.3 | 70.3 | 54.1 | 47.0 |
| Step-1000 | PRIOR | **1037.9** | **254.3** | **38.6** | **77.1** | **60.0** | **54.0** |
| | $\Delta$ | +90.9 | +12.7 | +1.3 | +6.8 | +5.9 | +7.0 |

## A  Theoretical Justification via Mutual Information

The efficacy of PRIOR can be further understood through the lens of mutual information theory. When optimizing vision-language models, we aim to maximize the predictive power of visual information $v$ with respect to text $t$. This can be formalized by maximizing the mutual information between $v$ and $t$, quantifying the uncertainty reduction about $t$ when $v$ is observed. For data sampled from the joint distribution $(v, t)$, the mutual information is expressed as:

$$I(v; t) = \mathbb{E}_{(v,t)} \left[ \ln \frac{p(v, t)}{p(v)p(t)} \right] = \mathbb{E}_{(v,t)} \left[ \ln \frac{p(t|v)}{p(t)} \right] \tag{13}$$

Decomposing this expression at the token level, we obtain:

$$I(v; t) = \mathbb{E}_{(v,t)} \left[ \sum_i \ln p(t_i|v, t_{<i}) - \ln p(t_i|t_{<i}) \right] \tag{14}$$

This formulation shows that mutual information is maximized when there's a large discrepancy between token probability given both visual and textual context $p(t_i|v, t_{<i})$ versus textual context alone $p(t_i|t_{<i})$. Tokens exhibiting this difference benefit most from visual inputs.

The weighting mechanism in PRIOR, defined as $w_i = (1 - p_r(t_i|t_{<i}))^\alpha$, implicitly aligns with this mutual information objective. By assigning higher importance scores to tokens that are difficult to predict from text alone, PRIOR effectively prioritizes tokens where visual information potentially provides the greatest reduction in uncertainty. This approach creates a natural emphasis on tokens where $\ln p(t_i|v, t_{<i}) - \ln p(t_i|t_{<i})$ is likely to be large, thus indirectly promoting higher mutual information between vision and language representations.

This intuitively corresponds to focusing loss optimization on the tokens where the LVLMs have the greatest opportunity to outperform text-only predictions by leveraging visual information. In essence, PRIOR adaptively modulates the learning signal based on the potential information gain from incorporating visual context, leading to more efficient and effective vision-language pre-training.

## B  Controlled Sensitivity Study over the Number of SFT Steps

In the main experiments (§3), all pre-trained checkpoints are evaluated after a lightweight 20-step instruction fine-tuning (SFT) stage. A natural question arises: does PRIOR's advantage originate from the pre-training

objective itself, or is it amplified by this particular fine-tuning regime? To answer this, we conduct a controlled sensitivity study by varying the number of SFT steps $\in \{20, 50, 500, 1000\}$ while keeping the SFT data, optimizer, and evaluation protocol fixed. All SFT runs start from the fully converged 5,000-step pre-trained H-LVLMs checkpoints. The results are presented in Tab. 2. We have the following observations: (1) The PRIOR–NTP gap persists across all SFT regimes. Across all six benchmarks and all four SFT configurations, PRIOR consistently outperforms NTP. This demonstrates that PRIOR's advantage is not an artifact of a particular fine-tuning duration. (2) PRIOR maintains advantages even with extended SFT. At Step-1000, PRIOR still outperforms NTP on all benchmarks (*e.g.,* +90.9 on MME-Perception, +6.8 on MMBench, +7.0 on SEEDBench), confirming that the gains are not merely amplified by limited fine-tuning but reflect genuinely improved representations learned during pre-training.

## C  Limitations of the Token Importance Heuristic

While PRIOR's text-only probability heuristic effectively identifies image-related tokens in the majority of cases, we acknowledge scenarios where low text-only probability does not correspond to visual relevance. We provide a qualitative analysis of such failure cases to better characterize the limitations of our approach.

**Rare Proper Nouns and Specific Identifiers**  Consider the following caption from our curated study dataset: "A hand-painted ceramic vase crafted by artisan Mikhail Vorontsov, displayed at the Hermitage Museum." In this case, visually verifiable tokens like "hand-painted" and "ceramic" receive moderate importance scores (*e.g.,* $w_i = 0.65$ and $0.55$, respectively), appropriately reflecting their image relevance. However, the artisan's first name "Mikhail" and the specific museum "Hermitage" receive very high importance scores ($w_i = 0.98$ and $0.92$) because they are rare and difficult to predict from textual context alone. These tokens represent metadata or provenance information that cannot be verified from the image, yet they receive disproportionately high weights in the PRIOR loss.

**Invisible Attributes and Non-Visual Metadata**  A related failure mode arises with attributes that are fundamentally non-visual. For example, given the caption "An authentic antique mahogany chair, appraised at \$47,500 and previously owned by aristocrats," the token "aristocrats" receives a high importance score ($w_i = 0.91$) due to its unpredictability from context. Similarly, "authentic" ($w_i = 0.62$) describes a property that requires expert verification rather than visual inspection. While visually informative tokens like "mahogany" ($w_i = 0.85$) are also appropriately upweighted, the heuristic cannot distinguish between tokens that are *hard to predict* because they describe visual content versus those that are hard to predict because they encode non-visual metadata.

**Numerical Specificity**  Exact quantities such as prices, dates, and measurements are inherently unpredictable from preceding text, leading to high importance scores. For instance, a price tag like "\$365,500" or a specific address like "7338 24th Ave NE" in a real estate caption will receive elevated importance scores despite being metadata that cannot be inferred from the associated image.

**Mitigating Factors**  Despite these cases, several aspects of PRIOR's design help mitigate their impact. First, the normalization term $w_i / \sum_{j=1}^{k} w_j$ bounds the maximum weight any single token can receive, preventing rare entities from dominating the loss. Second, these failure cases represent a relatively small fraction of all high-importance tokens in practice—the majority of tokens with low text-only probability genuinely correspond to visual content, as evidenced by the consistent performance improvements across all benchmarks.

