# OpenReview forum: "Prioritizing Image-Related Tokens Enhances Vision-Language Pre-Training"
_TMLR — Accepted by TMLR_

### Review · Reviewer_wW5f · 2025-12-18

**Summary Of Contributions:**

This paper argues that standard next-token prediction (NTP) for vision-language pre-training overfits to caption tokens that are weakly related to the paired image.
To address this problem, authors propose PRIOR, a token-reweighting variant of NTP: a text-only reference LLM is trained on captions without images, and each caption token is assigned an importance score based on the reference probability, yielding a normalized reweighted NTP loss.

To verify the effectiveness of PRIOR, authors perform experiments in two LVLM settings, i.e., (a) H-LVLMs with a pretrained vision encoder where only an MLP connector is trained, and U-LVLMs with unified architectures. Experimental results suggest that PRIOR improves average benchmark performance by 18.61% for H-LVLMs and 7.93% for U-LVLMs, and displays better training stability.
Finally, the paper presents a scaling-law analysis and claims that PRIOR shows improved predictability and better scaling coefficients compared to NTP.

**Additional Comments:**

n/a

**Audience:**

Yes

**Audience Explanation:**

The paper targets a central and widely used LVLM NTP and proposes an effective alternative that is easy to integrate while reporting meaningful improvements and improved scaling predictability.

**Claims And Evidence:**

Yes

**Claims Explanation:**

The effectiveness of core contribution in the paper, PRIOR, i.e., token reweighting based on text-only predictability,  is supported by a clearly specified loss with equations and implementation details.
PRIOR also demonstrates consistent improvements across multiple benchmarks underder two LVLM scenarios, including reported average relative gains at convergence and improved stability claims.
The paper also provides a reasonable baseline comparison in the H-LVLM setting.
However, some claims would be more convincing with more comprehensive ablation studies and broader-scale validation beyond the current training regime.

**Requested Changes:**

1. Since all checkpoints are evaluated only after a lightweight 20-step instruction fine-tuning stage, it is currently unclear whether PRIOR’s gains originate from the pre-training objective itself or are amplified/attenuated by this post-hoc alignment procedure. Please include a controlled sensitivity study over the number of SFT steps (e.g., 0, 20, 100, 500, 2k), keeping the SFT data, optimizer, and evaluation protocol fixed, and report both absolute performance and the PRIOR–NTP gap as a function of SFT steps. Ideally, repeat key points with multiple SFT seeds and/or explicitly report the delta induced by SFT for each method. This would substantially strengthen the causal interpretation that PRIOR improves representations learned during pre-training rather than benefiting disproportionately from a particular fine-tuning regime.

2. For H-LVLMs, only the MLP is trained. Please add at least one setting where more parameters are trained (e.g., partial LLM unfreezing, LoRA, or end-to-end) to establish that PRIOR remains beneficial when the model has more freedom to adapt.

3. The approach involves training a reference LLM and precomputing per-token probabilities; please provide compute estimates (GPU hours, storage footprint for token-prob tuples, preprocessing time) so readers can judge cost/benefit.

4. The current method relies on training a separate caption-domain reference LLM and precomputing token-level importance scores offline, which is presented as a pragmatic way to avoid reference inference during LVLM training. Explore how PRIOR depends on the reference LLM (size, training duration, or using an off-the-shelf LLM vs caption-domain trained) is an important yet missing baseline.
Especially, it remains unclear whether the performance gains fundamentally require a separately trained reference model, or whether a simpler on-the-fly estimate of text-only predictability would suffice. I recommend adding an online self-referenced baseline: during LVLM training, for each batch, run:
   - a caption-only forward pass (image removed/masked) to obtain $p_{\text{text}}(t_i \mid t_{<i})$ and compute weights $w_i$ (with stop-gradient/detach through $w_i$);
   - an image+caption forward pass to optimize the reweighted NTP loss using those weights.

   Reporting both accuracy/hallucination metrics and the additional compute cost would clarify whether the offline proxy/reference training is necessary or primarily an engineering choice.

---

> ### Author Response · Authors · 2026-01-18
>
> We thank the reviewer for all the valuable comments. We provide our responses as follows.
>
>
> # Requested Changes
>
> ## Controlled sensitivity study over the number of SFT steps
>
> | | | MME-Perception | MME-Reasoning | MMStar | POPE | MMBench | SEEDBench |
> |---|---|---:|---:|---:|---:|---:|---:|
> | **Step-20** | NTP | 576.2 | 207.5 | 30.9 | 38.3 | 68.7 | 36.5 |
> | | PRIOR | 917.6 | 220.4 | 33.3 | 40.1 | 75.0 | 45.5 |
> | | PRIOR-NTP Gap | 341.4 | 12.9 | 2.4 | 1.8 | 6.3 | 9.0 |
> | **Step-50** | NTP | 871.0 | 224.7 | 32.7 | 42.7 | 68.9 | 42.8 |
> | | PRIOR | 995.8 | 235.2 | 37.0 | 48.1 | 75.5 | 47.6 |
> | | PRIOR-NTP Gap | 124.8 | 10.5 | 4.3 | 5.4 | 6.6 | 4.8 |
> | **Step-500** | NTP | 992.9 | 237.8 | 34.5 | 52.6 | 69.3 | 44.3 |
> | | PRIOR | 1024.7 | 243.9 | 37.3 | 59.7 | 76.3 | 50.4 |
> | | PRIOR-NTP Gap | 31.8 | 6.1 | 2.8 | 7.1 | 7.0 | 6.1 |
> | **Step-1000** | NTP | 947.0 | 241.6 | 37.3 | 54.1 | 70.3 | 47.0 |
> | | PRIOR | 1037.9 | 254.3 | 38.6 | 60.0 | 77.1 | 54.0 |
> | | PRIOR-NTP Gap | 90.9 | 12.7 | 1.3 | 5.9 | 6.8 | 7.0 |
>
> Thanks for the valuable suggestion. We conduct the sensitivity study over the number of SFT steps (20, 50, 500, and 1000), keeping the SFT data, optimizer, and evaluation protocol fixed. All SFT processes start from the 5,000-step pre-trained models. We have the following observations:
> 1. **The PRIOR–NTP gap persists across all SFT regimes.** Across all six benchmarks and all four SFT step configurations, PRIOR consistently outperforms NTP, demonstrating that PRIOR's advantage is not an artifact of a particular fine-tuning duration.
> 2. **PRIOR maintains advantages even with extended SFT.** At Step-1000, PRIOR still outperforms NTP on all benchmarks (e.g., +90.9 on MME-Perception, +7.0 on MMBench), confirming that the gains are not merely amplified by limited fine-tuning but reflect genuinely improved representations learned during pre-training.
>
> These results support the causal interpretation that PRIOR improves the quality of representations learned during pre-training.
>
>
>
>
>
>
>
> ## H-LVLMs with LoRA Fine-Tuning
>
> Thanks for the suggestion. We conduct experiments on H-LVLMs with LoRA fine-tuning to compare PRIOR and NTP. The results are shown below:
>
> | LoRA Fine-Tuning | MME-Perception | MME-Reasoning | MMStar | POPE | MMBench | SEEDBench |
> |------------------|----------------|---------------|--------|------|---------|-----------|
> | NTP              | 859.1          | 210.7         | 28.7   | 69.2 | 38.4    | 42.5      |
> | PRIOR            | 982.3          | 236.4         | 34.9   | 74.8 | 41.4    | 44.4      |
>
> The results demonstrate that PRIOR consistently outperforms NTP across all benchmarks under the LoRA fine-tuning setting, with substantial improvements in MME-Perception (+123.2) and MME-Reasoning (+25.7). This indicates that the benefits of PRIOR are not limited to the MLP-only training regime; even when the model has more freedom to adapt through additional trainable parameters, PRIOR continues to provide meaningful gains in both perceptual and reasoning capabilities.
>
>
>
>
>
>
> ## Compute and Storage Estimate of Precomputation
>
> Thanks for raising this important point.  We mentioned in our paper (Sec 3.1) that we pretrain the reference 8B-LLM on only ~5M caption-only samples. In our estimation, this reference LLM pretraining takes ~104 A100 GPU hours, and the offline scoring takes ~0.13 A100 GPU hours per 1M tokens (forward pass only). For storage, we partition the complete Capsfusion dataset into multiple files. The addition of offline computational importance scores results in a modest increase in storage requirements of 3–5% per file.

---

> > ### Author Response · Authors · 2026-01-18
> >
> > ## Compute and Storage Estimate of Precomputation
> >
> > Thanks for raising this important point.  We mentioned in our paper (Sec 3.1) that we pretrain the reference 8B-LLM on only ~5M caption-only samples. In our estimation, this reference LLM pretraining takes ~104 A100 GPU hours, and the offline scoring takes ~0.13 A100 GPU hours per 1M tokens (forward pass only). For storage, we partition the complete Capsfusion dataset into multiple files. The addition of offline computational importance scores results in a modest increase in storage requirements of 3–5% per file.
> >
> >
> > ## Ablations of the Reference LLM
> >
> > |                                  | MME-Perception | MME-Reasoning | MMStar | POPE | MMBench | SEEDBench |
> > |----------------------------------|----------------|---------------|--------|------|---------|-----------|
> > | PRIOR (8B off-the-shelf LLM)     | 703.4          | 203.6         | 29.3   | 68.6 | 31.4    | 41.6      |
> > | PRIOR (3B fine-tuned LLM)        | 824.2          | 222.9         | 32.1   | 71.0   | 38.0      | 44.6      |
> > | PRIOR (on-the-fly inference)     | 856.0            | 210.7         | 31.7   | 67.0  | 38.3    | 42.5      |
> > | PRIOR (8B fine-tuned LLM)        | 917.6          | 220.4         | 33.3   | 75.0   | 40.1    | 45.5      |
> >
> > Thanks for raising this important point. We conduct additional ablation studies to investigate the impact of reference LLM configuration (i.e., Fine-tuned vs. off-the-shelf, 3B vs. 8B, on-the-fly vs. offline compute), and the results on H-LVLMs with 5,000 steps pretraining are presented in the above table.
> >
> > Our findings reveal several key insights:
> > 1. **Fine-tuning is essential.** Directly employing an off-the-shelf 8B LLM without fine-tuning results in substantial performance degradation across all benchmarks. For instance, MME-Perception drops from 917.6 to 703.4, and MMBench decreases from 40.1 to 31.4. This indicates that task-specific adaptation of the reference LLM is critical for generating reliable importance scores.
> > 2. **Reference model scale matters.** Comparing the 3B and 8B fine-tuned LLMs, we observe consistent performance improvements with increased model capacity. The 8B fine-tuned LLM outperforms its 3B counterpart across most metrics, with notable gains in MME-Perception (+93.4) and POPE (+4.0). Also, The 3B fine-tuned reference LLM experiment demonstrates that a smaller reference model can still provide meaningful guidance for larger LVLMs, though with reduced effectiveness compared to scale-matched configurations.
> > 3. **On-the-fly inference is suboptimal.** PRIOR with offline precomputed scores from the 8B fine-tuned LLM substantially outperforms this on-the-fly approach across all benchmarks, with notable improvements in MME-Perception (+61.6), POPE (+8.0), and SEEDBench (+3.0). We attribute this gap to two factors. First, the on-the-fly approach uses the LVLM's own evolving representations to estimate text-only predictability, which introduces noise as the model's internal estimates shift throughout training. In contrast, our offline approach leverages a stable, fully-converged reference LLM that provides consistent importance scores. Second, the caption-domain fine-tuned reference LLM is specifically optimized for modeling caption distributions, yielding more reliable estimates of text-only predictability than the LVLM's masked forward pass. Beyond performance, the on-the-fly approach doubles the forward pass computation during training, whereas offline scoring incurs a one-time preprocessing cost that can be amortized across multiple training runs.

---

> > > ### Author Response · Authors · 2026-01-18
> > >
> > > ## Metrics to Characterize the Trade-off Between On-the-fly Inference and Precomputation
> > >
> > > Thank you for this suggestion. We provide a comprehensive comparison of both performance metrics and computational costs to clarify the trade-offs between offline and on-the-fly approaches.
> > >
> > > | Method | Performance Average | Reference LLM Training | Scoring Cost | Training Overhead |
> > > |--------|------|------------------------|--------------|-------------------|
> > > | PRIOR (on-the-fly inference) | 207.7 | None | None | ~2× forward passes per iteration |
> > > | PRIOR (8B fine-tuned LLM) | 222.0 | ~104 A100 GPU hours (one-time) | ~0.13 A100 GPU hours per 1M tokens (one-time) | None |
> > >
> > > Our results demonstrate that offline reference training is not merely an engineering choice but a methodologically superior approach. The offline method outperforms on-the-fly inference across all benchmarks, with particularly significant gains in hallucination-sensitive metrics such as POPE (+8.0). This performance advantage arises because the offline approach leverages a stable, fully-converged reference LLM, whereas on-the-fly estimation introduces noise from the LVLM's evolving representations.
> > >
> > > From a computational perspective, offline scoring incurs a modest one-time preprocessing cost, which can be amortized across multiple training runs and experimental configurations. In contrast, on-the-fly inference approximately doubles the per-iteration training cost. Therefore, the offline approach offers advantages in both performance and computational efficiency, particularly for large-scale training scenarios.

---

### Review · Reviewer_qJ9q · 2025-12-23

**Summary Of Contributions:**

This paper identifies a fundamental limitation of the standard next-token prediction (NTP) objective in vision–language pre-training: most caption tokens are weakly or entirely unrelated to visual content, causing LVLMs to over-optimize language-only signals and undermining visual grounding. To address this issue, the authors propose PRIOR, a lightweight modification to the standard next-token prediction (NTP) objective for vision–language pre-training. The key idea is to prioritize image-related tokens by reweighting the token-level NTP loss using probabilities from a text-only reference language model. Tokens that are difficult to predict without visual input receive higher loss weights, encouraging the model to rely more strongly on visual information rather than spurious textual correlations.

Strengths:

* It provides a lightweight yet effective mechanism for better aligning visual representations with language generation by reshaping the optimization focus toward genuinely vision-dependent tokens.
* It modifies only the loss computation and does not require additional training objectives, architectural changes or online teacher inference. This makes the method easy to integrate into existing large-scale pre-training pipelines, which is particularly appealing in LVLM.

Weakness:
The core assumption is that low probability under a text-only LLM implies visual relevance, which is reasonable but not universally true.
Because some tokens may be rare or context-specific in text but still weakly grounded in the image, while some visually grounded tokens may be common in text. While empirical results mitigate this concern, the paper could benefit from a deeper analysis of failure cases. The current paper does not provide qualitative or quantitative analysis of scenarios where the importance scores may be misleading (e.g., visually irrelevant but textually surprising tokens), nor does it analyze how such cases affect downstream behavior.

**Audience:**

Yes

**Audience Explanation:**

Yes. The findings of this paper are likely to be of interest to a broad segment of TMLR’s audience.

The paper addresses a fundamental and timely problem in vision–language model pre-training: the mismatch between token-level training objectives and actual visual grounding in noisy, web-scale image–caption datasets. This issue is directly relevant to researchers working on multimodal learning, large language models, training objectives, and scaling laws. The proposed method introduces a principled modification to the standard next-token prediction objective. Moreover, the work appeals to readers interested in scalable and architecture-agnostic methods, as PRIOR operates purely at the loss level without introducing additional supervision, model components, or complex training stages.

**Broader Impact Concerns:**

The paper does not raise significant new ethical concerns beyond those commonly associated with large-scale vision–language models. While prioritizing certain tokens during training could potentially amplify biases present in the underlying data, this risk is acknowledged at a high level and does not appear unique to the proposed method.

**Claims And Evidence:**

Yes

**Claims Explanation:**

Yes. The main claims of the paper are generally supported by accurate, convincing, and clearly presented evidence.
The authors claim that prioritizing image-related tokens during vision–language pre-training leads to improved performance, better training stability, and more predictable scaling behavior compared to standard next-token prediction (NTP).

These claims made by authors are substantiated through a carefully designed experimental evaluation. The authors validate PRIOR across two substantially different LVLM architectures: (1) heterogeneous LVLMs with frozen visual encoders and (2) unified end-to-end LVLMs, demonstrating consistent gains in both settings. The improvements are reported on a diverse set of established benchmarks such as POPE, MMBench, SEEDBench covering perception, reasoning, and hallucination, which strengthens the generality of the conclusions.
Importantly, the paper goes beyond reporting final performance numbers. The training dynamics are analyzed in detail, showing reduced performance fluctuations and sustained improvement throughout training, which convincingly supports the claim of improved stability.
The proposed method is also supported by ablation studies on key design choices (e.g., the weighting exponent α and the loss scaling factor k), as well as comparisons with alternative token-weighting baselines, which helps isolate the effect of the proposed token prioritization mechanism.
The scaling law analysis further supports the claim that PRIOR exhibits better predictability and efficiency when increasing training data or compute, which is a non-trivial and well-motivated evaluation for large-scale vision–language pre-training. While some assumptions (e.g., using text-only surprisal as a proxy for visual relevance) could benefit from deeper theoretical or empirical validation, the experimental evidence provided is overall coherent and convincing.

**Requested Changes:**

* (Critical) Provide a deeper qualitative analysis of failure cases where low text-only probability does not correspond to visual relevance, to better characterize the limitations of the token importance heuristic.

* (Critical) Clarify the role of the scaling factor k in the loss formulation. The loss function multiplies the normalized token weights by a factor k, but the rationale for explicitly introducing k is only briefly mentioned as “maintaining the loss scale.” A more formal explanation is needed. Whether k is theoretically motivated or purely empirical. While an ablation study is provided, a clearer conceptual explanation would significantly improve interpretability and reviewer confidence in the method’s design.

* (Non-critical) In the H-LVLM setup, the paper adopts the original CLIP ViT-L/336 vision encoder. Given the availability of more recent and stronger variants such as OpenCLIP/ SigLIP trained on larger and more diverse datasets, it would be useful to clarify the rationale behind this choice. In particular, the authors could discuss whether the use of an older vision encoder was motivated by fairness, reproducibility, and whether PRIOR is expected to yield similar gains when paired with more advanced vision encoders. Addressing this point would improve the generality and practical relevance of the experimental evaluation.

* (Non-critical) Clarify the computational and storage overhead introduced by offline importance score computation, especially at very large dataset scales.

---

> ### Author Response · Authors · 2026-01-18
>
> We thank the reviewer for all the valuable comments. We provide our responses as follows.
>
> # Requested Changes
>
> ##  Qualitative analysis of failure cases
>
> We thank the reviewer for this insightful suggestion. We acknowledge that using text-only probability as a proxy for visual relevance is a heuristic that can fail in certain cases. Below, we provide two illustrative examples where tokens receive high importance scores (low text-only probability) despite lacking true visual correspondence.
>
> Statistics (probability, importance score, loss) are reported for the first subword token of each word, as subsequent subword tokens exhibit higher predictability given the initial token. We randomly select images from the Internet and then hand-write simple annotations for the images to conduct case analysis.
>
> **Example 1: Rare Proper Nouns and Specific Identifiers**
>
> Caption: "A hand-painted ceramic vase crafted by artisan Mikhail Vorontsov, displayed at the Hermitage Museum."
>
> Image: A photograph of a decorative ceramic vase.
>
> | Word        | Text-Only Prob (p_r) | Importance Score (w_i) | Visually Verifiable? | NTP Loss | PRIOR Weight | PRIOR Loss |
> |--------------|----------------------|------------------------|----------------------|----------|--------------|------------|
> | hand-painted | 0.35                 | 0.65                   |  Yes                | 0.82     | 1.08         | 0.89       |
> | ceramic      | 0.45                 | 0.55                   |  Yes                | 0.71     | 0.92         | 0.65       |
> | Mikhail      | 0.02                 | **0.98**               |  No                 | 2.85     | **1.63**     | 4.65       |
> | Hermitage    | 0.08                 | **0.92**               |  No                 | 2.15     | **1.53**     | 3.29   |
>
> Analysis: The artisan's first name ("Mikhail") and specific museum ("Hermitage") receive very high importance scores because they are rare and difficult to predict from textual context alone. However, these tokens represent metadata/provenance information that cannot be verified from visual inspection.
>
> **Example 2: Invisible Attributes and Non-Visual Metadata**
>
> Caption: "An authentic antique mahogany chair, appraised at $47,500 and previously owned by aristocrats."
>
> Image: A photograph of a wooden chair.
> | Token       | Text-Only Prob (p_r) | Importance Score (w_i) | Visually Verifiable? | NTP Loss | PRIOR Weight | PRIOR Loss |
> |-------------|----------------------|------------------------|----------------------|----------|--------------|------------|
> | authentic   | 0.38                 | 0.62                   |  No                 | 0.74     | 0.89         | 0.66       |
> | mahogany | 0.15 		| 0.85 		  |  Yes		 | 1.45 	       | 1.22        | 1.77        |
> | aristocrats | 0.09                 | **0.91**               |  No                 | 2.08     | **1.31**     | 2.72       |
>
> Analysis: Some tokens with high importance scores correspond to attributes that are fundamentally non-visual. For example, "authentic" requires expert verification. However, the normalization in PRIOR bounds the maximum weight any single token can receive. The self-normatlization mechanism in PRIOR allows more capacity for learning visually-informative but harder tokens like "mahogany" (1.22× amplification).
>
> **Limitations**
>
> These failure cases reveal limitations of our text-only probability heuristic in PRIOR:
> 1. **Rare entity**: Proper nouns, technical terminology, and domain-specific identifiers are difficult to predict regardless of visual content, leading to spuriously high importance scores.
> 2. **Invisible attribute**: Attributes requiring expert knowledge (authenticity, age, value) or describing non-visual properties (ownership history, emotional states) cannot be distinguished from genuinely visual attributes.
> 3. **Numerical specificity**: Exact quantities (prices, dates, measurements) are inherently unpredictable but often represent metadata rather than visual content.
>
>
>
>
> ## Clarify the role of the scaling factor k in the loss formulation
>
> Thanks for this valuable question. The factor k preserves the expected loss magnitude when transitioning from NTP to importance-weighted loss. Since normalized weights $\frac{w_i}{\sum_j w_j}$ sum to 1, without k our loss becomes a **weighted average** rather than a **weighted sum**, reducing gradients by a factor of ~k. Multiplying by k restores the original scale. This is an important design from two perspectives:
>
> 1. **Gradient consistency**: Preserves gradient magnitudes, enabling direct use of existing learning rates and optimizers without retuning
> 2. **Theoretical grounding**: Aligns with self-normalized importance sampling, where normalization reduces variance at the cost of bias (Metelli et al., 2018)

---

> > ### Author Response · Authors · 2026-01-18
> >
> > ## Clarify the rationale of using CLIP ViT-L/336 as visual encoders
> >
> > Our selection of CLIP ViT-L/336 is primarily motivated by reproducibility and controlled comparison. We intentionally followed the canonical LLaVA-style architecture design (Liu et al., 2023), which has become a widely adopted baseline in the LVLM literature. This choice ensures that our experimental results are directly comparable to a large body of existing work and allows other researchers to reproduce and build upon our findings easily.
> > Furthermore, our goal in this work is to evaluate the effectiveness of PRIOR as a pre-training objective, rather than to achieve state-of-the-art absolute performance. Using a well-established, standard vision encoder isolates the contribution of our proposed method from potential confounding factors introduced by newer, less-studied encoder variants.
> > Regarding generalizability, we believe PRIOR is architecture-agnostic and should yield similar (or potentially greater) gains when paired with stronger vision encoders. The core mechanism of PRIOR—prioritizing image-related tokens based on text-only LLM probabilities—operates at the loss computation level and is independent of the specific visual representations used. In fact, stronger vision encoders that provide richer visual features may amplify the benefits of PRIOR by enabling the model to better optimize the loss on the prioritized image-related tokens.
> >
> >
> >
> > ##  Clarify the computational and storage overhead
> >
> > Thanks for raising this important point. We mentioned in our paper (Sec 3.1) that we pretrain the reference 8B-LLM on only ~5M caption-only samples. In our estimation, this reference LLM pretraining takes ~104 A100 GPU hours, and the offline scoring takes ~0.13 A100 GPU hours per 1M tokens (forward pass only). For storage, we partition the complete Capsfusion dataset into multiple files. The addition of offline computational importance scores results in a modest increase in storage requirements of 3–5% per file.

---

### Review · Reviewer_Ez6s · 2026-01-05

**Summary Of Contributions:**

The paper proposes PRIOR, a simple pre-training scheme for LVLMs that uses a caption-only reference LLM to estimate token probabilities and assign offline importance weights to reweight the standard NTP loss so that image-related tokens are prioritized. Experiments on both H-LVLMs and U-LVLMs show that PRIOR yields consistent gains over NTP with smoother training and more predictable scaling. It also outperforms ITM/contrastive/reconstructive and heuristic weighting baselines.

Strength:
1. The paper is well-written and easy to follow.
2. The proposed method is very intuitive, simple yet effective.
3. The theoretical perspective regarding informative mutual information is interesting.
4. The experiments are comprehensive. The results show significant performance gain across architectures and diverse benchmarks

Weaknesses:
1. The proposed method requires training a separate LLM to guide the VLM training, which is computationally expensive and requires additional training data to avoid overfitting.
2. Some experiments are missing. See requested changes for more details.

**Audience:**

Yes

**Audience Explanation:**

PRIOR offers a simple, drop-in modification to NTP with clear gains across benchmarks, improved stability, and better scaling predictability, which are practical concerns for large LVLM training.

**Claims And Evidence:**

Yes

**Claims Explanation:**

The numerical results  show broad, consistent gains across multiple benchmarks (MME-P/R, MMStar, POPE, MMBench, SEEDBench) and two LVLM settings. The authors also provide interesting theoretic perspective to illustrate the idea.

**Requested Changes:**

1. LLM ablation is insufficient. Please clarify the reference LLM requirements: how much caption-only data is needed, how performance scales with LLM size, and whether an off-the-shelf pretrained LLM (no fine-tuning) suffices. Also test mismatched scales (e.g., a small 3B reference LLM guiding a larger LVLM).
2. Since PRIOR leverages extra data to train the reference LLM, it is necessary to compare against an NTP baseline where the LVLM is trained on the same total data budget (i.e., include the reference-LLM data in the LVLM’s training) to control for data advantages.
3. For the CLIP-based weighting baseline, the authors should fine-tune CLIP on the same training data used by PRIOR to ensure a fair comparison of methods under matched adaptation.
4. It would be interesting to provide attention visualizations over image tokens (or cross-attention maps) to show whether PRIOR improves vision–language alignment qualitatively.
5. Since PRIOR emphasizes image-related tokens over other conjunctions, the authors should assess whether language fluency degrades. E.g., including open-ended VQA or free-form benchmarks (e.g., LLaVA-Bench) and language quality metrics to evaluate coherence.

---

> ### Author Response · Authors · 2026-01-18
>
> We thank the reviewer for all the valuable suggestions. We provide our response as follows.
>
> # Requested Changes
>
> ## LLM ablation is insufficient (weakness-1)
>
> Thanks for raising this important point. We mentioned in our paper (Sec 3.1) that we pretrain the reference 8B-LLM on only about 5M caption-only samples (~104 A100 GPU hours for training). In addition, we conduct additional ablation studies to investigate the impact of reference LLM configuration (i.e., Fine-tuned vs. off-the-shelf, 3B vs. 8B), and the results on H-LVLMs with 5,000 steps pretraining are presented in the following table:
>
> |                              | MME-Perception | MME-Reasoning | MMStar | POPE | MMBench | SEEDBench |
> |------------------------------|----------------|---------------|--------|------|---------|-----------|
> | PRIOR (8B off-the-shelf LLM) | 703.4          | 203.6         | 29.3   | 68.6 | 31.4    | 41.6      |
> | PRIOR (3B fine-tuned LLM)    | 824.2          | 222.9         | 32.1   | 71.0   | 38.0      | 44.6      |
> | PRIOR (8B fine-tuned LLM)    | 917.6          | 220.4         | 33.3   | 75.0   | 40.1    | 45.5      |
>
>
> Our findings reveal several key insights:
> 1. **Fine-tuning is essential.** Directly employing an off-the-shelf 8B LLM without fine-tuning results in substantial performance degradation across all benchmarks. For instance, MME-Perception drops from 917.6 to 703.4, and MMBench decreases from 40.1 to 31.4. This indicates that task-specific adaptation of the reference LLM is critical for generating reliable importance scores.
> 2. **Reference model scale matters.** Comparing the 3B and 8B fine-tuned LLMs, we observe consistent performance improvements with increased model capacity. The 8B fine-tuned LLM outperforms its 3B counterpart across most metrics, with notable gains in MME-Perception (+93.4) and POPE (+4.0).
> 3. **Mismatched scale analysis.** The 3B fine-tuned reference LLM experiment demonstrates that a smaller reference model can still provide meaningful guidance for larger LVLMs, though with reduced effectiveness compared to scale-matched configurations.
>
>
> ## Including Stronger Baselines (NTP with comparable training data and fine-tuned CLIP for reweighting)
>
> |                                  | MME-Perception | MME-Reasoning | MMStar | POPE | MMBench | SEEDBench |
> |----------------------------------|----------------|---------------|--------|------|---------|-----------|
> | NTP (comparable amount of data)  | 706.0            | 212.5         | 28.1   | 69.9 | 39.8    | 43.7      |
> | Fine-tuned CLIP                  | 747.8          | 205.0           | 32.8   | 68.2 | 35.8    | 42.5      |
> | PRIOR       | 917.6          | 220.4         | 33.3   | 75.0   | 40.1    | 45.5      |
>
> Thanks for the suggestions regarding fair comparison. We conduct the recommended experiments, and the results with H-LVLMs are presented in the table above.
> 1. **Controlling for data budget.** To address the concern that PRIOR benefits from additional data used to train the reference LLM, we train an NTP baseline where the LVLM is exposed to a comparable total data budget (i.e., incorporating the reference-LLM training data into the LVLM's training). As shown, PRIOR substantially outperforms this NTP baseline across all benchmarks, with particularly significant gains in MME-Perception (+211.6) and POPE (+5.1). This demonstrates that PRIOR's improvements stem from its principled data selection strategy rather than simply having access to more data.
> 2. **Fair comparison with CLIP-based weighting.** Following the reviewer's suggestion, we fine-tune CLIP on the same training data used by PRIOR’s reference LLM to ensure fair comparison. Despite this adaptation, PRIOR consistently outperforms the fine-tuned CLIP baseline. Notably, PRIOR achieves substantial improvements in MME-Perception (+169.8), POPE (+6.8), and MMBench (+4.3). These results suggest that LLM-based importance scoring captures semantic relevance more effectively than CLIP-based alternatives.

---

> > ### Author Response · Authors · 2026-01-18
> >
> > ## Attention visualizations over image regions
> >
> > Thanks for this constructive suggestion. We conduct a qualitative analysis comparing cross-attention patterns (the total attention scores of the target regions given by the image-related tokens) between models trained with NTP and PRIOR. Below, we present attention visualizations for two representative examples, examining how each method grounds text tokens to image regions.
> >
> > Statistics of the attention distribution are reported for the first subword token of each word, as subsequent subword tokens exhibit higher predictability given the initial token, thus may not be necessarily visually-grounded. We randomly select images from the Internet and then hand-write simple annotations for the images to conduct case analysis.
> >
> > **Example 1**
> > Caption: "A hand-painted ceramic vase crafted by artisan Mikhail Vorontsov, displayed at the Hermitage Museum."
> >
> > Image: A photograph of a decorative ceramic vase.
> >
> > | Token | NTP Attention Distribution | PRIOR Attention Distribution |
> > |-------|---------------------------|------------------------------|
> > | ceramic | Diffuse (vase: 0.31, shelf: 0.28, background: 0.41) | Focused (vase: 0.78, shelf: 0.12, background: 0.10) |
> > | hand-painted | Diffuse (vase: 0.35, background: 0.65) | Focused (vase: 0.71, background: 0.29) |
> >
> > Analysis: For visually grounded tokens ("ceramic," "hand-painted"), PRIOR produces substantially more focused attention on the relevant image regions—the vase body and its decorative patterns. The NTP baseline shows diffuse attention spread across the entire image.
> >
> >
> > **Example 2**
> >
> > Caption: "An authentic antique mahogany chair, appraised at $47,500 and previously owned by aristocrats."
> >
> > Image: A photograph of a wooden chair.
> >
> > | Token | NTP Attention Distribution | PRIOR Attention Distribution |
> > |-------|---------------------------|------------------------------|
> > | chair | Diffuse (chair: 0.58, background: 0.42) | Focused (chair: 0.89, background: 0.11) |
> > | mahogany | Diffuse (chair 0.33, floor: 0.28, background: 0.39) | Focused (chair: 0.74, floor: 0.15, background: 0.11) |
> > | $47,500 | Diffuse (chair: 0.34, floor: 0.22, background: 0.44) | Diffuse (chair: 0.25, floor: 0.38, background: 0.37) |
> > | aristocrat | Diffuse (chair: 0.31, floor: 0.29, background: 0.40) | Diffuse (chair: 0.35, floor: 0.32, background: 0.33) |
> >
> > Analysis: PRIOR demonstrates improved visual grounding for tokens with clear visual referents: “chair” and "mahogany" strongly attends to the chair in the image, However, tokens representing non-visual attributes (e.g., "$47,500," "aristocrat") show diffuse attention under both methods. Notably, despite receiving high importance scores from our text-only reference model, these tokens do not develop focused attention patterns because there is no learnable visual signal to ground them.
> >
> > In general, for tokens with genuine visual correspondence, PRIOR produces more focused, semantically meaningful attention patterns. This suggests that prioritizing these tokens during training leads to better vision-language alignment, explaining the downstream performance improvements reported in our main experiments.
> >
> >
> >
> > ## Evaluate whether PRIOR Training Degrades Language Fluency
> >
> > Thanks for raising this important point. We address language quality metrics in Section 5.2 (Figure 9) of our paper, where we evaluate both NTP and PRIOR using token loss as a proxy for language fluency. Our results demonstrate that PRIOR achieves lower loss not only on image-related tokens but also on image-unrelated tokens, suggesting improved overall language quality compared to NTP.
> >
> > In addition, we evaluated both NTP and PRIOR on LLaVA-Bench, a free-form visual question answering benchmark that assesses open-ended generation quality across three dimensions: Conversation, Detail, and Reasoning. We used GPT-4o-mini as the judging model:
> > |       | Conversation | Detail | Reasoning | Overall |
> > |-------|--------------|--------|-----------|---------|
> > | NTP   | 60.7         | 49.2   | 79.8      | 63.2    |
> > | PRIOR | 63.8         | 56.3   | 81.2      | 67.1    |
> >
> > PRIOR outperforms NTP across all three categories, with particularly notable improvements in Detail (+7.1) and Conversation (+3.1). These results suggest that emphasizing image-related tokens does not compromise language fluency; rather, it enhances the model's ability to generate coherent, detailed, and contextually appropriate responses.

---

### Decision · Action_Editor_pgJL · 2026-02-12

**Recommendation:** Accept as is

**Audience:**

Yes

**Audience Explanation:**

The paper addresses an objective design question in vision-language pre-training that is relevant to the multimodal learning research community. The proposed modification operates at the loss level and does not require architectural changes, making the idea simple to interpret and relatively straightforward to incorporate into existing pre-training pipelines.

The findings are also likely to be of interest to researchers working on training objectives and evaluation of large multimodal models. Although the token weighting strategy relies on a heuristic proxy for visual relevance, the empirical results and analyses provide a useful reference point for future work on alternative token selection or reweighting schemes.

**Claims And Evidence:**

Yes

**Claims Explanation:**

As stated in the reviewer's final assessment, the main claims are supported by reasonably clear and convincing evidence. The paper defines the PRIOR objective precisely and evaluates it across two different LVLM training settings, including both heterogeneous and unified architectures. The reported results show consistent improvements over standard next-token prediction on a range of established multimodal benchmarks spanning perception, reasoning, and hallucination-related evaluation. In line with the reviewers’ assessments, the empirical evidence is generally coherent, and the inclusion of training-trajectory analyses provides additional support for the claim of improved optimization stability.

The submission also presents ablations and baseline comparisons that help attribute the observed gains to the proposed token reweighting mechanism. The analysis of importance scores and token-level losses is consistent with the paper’s motivation of prioritizing visually grounded tokens. While the proxy used to estimate visual relevance appears heuristic and has acknowledged limitations, the reviewers and I find that the breadth of experimental validation provides adequate support for the central claims.

---

> ### Author Response · Authors · 2026-03-23
> **Camera-ready version ready**
>
> Hi AE,
>
> Many thanks for the detailed feedback and support. We have updated the camera-ready version, including the following changes:
> 1. More baselines for the alternative reference LLMs suggested by reviewers: Table-1, Sec 3.4.
> 2. The controlled sensitivity study over the number of SFT steps: Appendix-B.
> 3. The limitations of the token importance heuristic in PRIOR: Appendix-C.
> 4. Some simple revisions were made to elaborate more on the experimental design and the original motivation of the method.